# Outdoor Time in Childhood: A Mixed Methods Approach to Identify Barriers and Opportunities for Intervention in a Racially and Ethnically Mixed Population

**DOI:** 10.3390/ijerph20247149

**Published:** 2023-12-06

**Authors:** Magdalena K. Haakenstad, Maria B. Butcher, Carolyn J. Noonan, Amber L. Fyfe-Johnson

**Affiliations:** 1Institute for Research and Education to Advance Community Health, Elson S. Floyd College of Medicine, Washington State University, Seattle, WA 98101, USA; m.haakenstad@wsu.edu (M.K.H.); carolyn.noonan@wsu.edu (C.J.N.); 2Elson S. Floyd College of Medicine, Washington State University, Spokane, WA 99202, USA; maria.butiu@wsu.edu

**Keywords:** outdoor time in childhood, barriers to outdoors, facilitators for outdoors, cultural perspectives on outdoor time, people of color, social determinants of health, mixed methods, nature, early childhood education

## Abstract

A growing body of literature suggests that outdoor time is beneficial for physical and mental health in childhood. Profound disparities exist in access to outdoor spaces (and the health benefits thereof) for children in communities of color. The objectives of this research were to: (1) identify challenges and solutions to outdoor time for children; (2) assess the importance of outdoor time for children; and (3) evaluate results stratified by race/ethnicity. Using a convergent mixed methods approach, we conducted a thematic analysis from 14 focus groups (*n* = 50) with outdoor educators, parents with children attending outdoor preschools, and community members with children. In addition, 49 participants completed a survey to identify challenges and solutions, perceived importance, and culturally relevant perspectives of outdoor time. The main challenges identified for outdoor time were safety concerns, inclement weather, lack of access to outdoor spaces, and parent work schedules. The primary proposed solution was integrating outdoor time into the school day. Nearly all participants, independent of racial identity, reported that outdoor time improved physical and mental health. Overall outdoor time was lower in participants from communities of color (~8 h/week) compared to their White counterparts (~10 h/week). While 50% of people of color (POC) reported that outdoor time was an important cultural value, only 18% reported that people in their respective culture spent time outside. This work contributes to accumulating knowledge that unique barriers to outdoor time exist for communities of color, and the children that live, learn, and play in these communities. Increasing outdoor time in school settings offers a potential solution to reduce identified barriers and to promote health equity in childhood.

## 1. Introduction

Nature contact is known to improve children’s health [1,2,3,4,5] by increasing physical activity [6,7], improving mental health [8,9,10,11], reducing stress [12,13], and promoting behavioral health [14,15,16]. Communities of color have disproportionately lower access to outdoor spaces [17,18,19,20] and experience more environmental barriers in outdoor spaces than their White counterparts [21,22,23,24,25,26]. Common environmental barriers in communities of color are safety concerns [21,22,23,24,25,26], lack of facilities [26,27,28], and lack of sidewalks [21,22,24,29]. One study found that 74% of communities of color in the United States (US) live in nature-deprived areas, compared with just 23% of White communities [30]. Another study examining park equity in Los Angeles, California, found that neighborhoods inhabited predominantly (>75%) by Latinos, African Americans, and Asian-Pacific Islanders had dramatically lower access to parks (0.6, 1.7, 0.3 park acres per 1000 population, respectively) than largely White areas of the city (31.8 park acres per 1000 population) [31]. Given that nature contact in childhood has known health benefits [1,6,9], yet communities of color disproportionately live, work, and play in nature-deprived areas, efforts are needed to increase access to nature-rich spaces in these communities, and the children therein.

Exposure to nature-rich spaces necessitates that these spaces exist and are accessible. To ensure that future policy-level decisions around outdoor spaces in communities of color are relevant, it is crucial to better understand the specific barriers and challenges that communities of color, and families within these communities, face. To date, most of the research in this area has focused on adult populations [22,23,26] or barriers to physical activity independent of outdoor time [26,27,28]. Only three studies have explored barriers to and facilitators for children’s outdoor active play [32,33,34]. While informative, none of these studies: (1) identified barriers to outdoor time in communities of color, or (2) used a mixed methods approach to assessing barriers to outdoor time for children. One study was a scoping review that emphasized the importance of outdoor play but did not directly assess barriers and challenges to outdoor play in early childhood [33]. Another used parent and clinician-based focus groups to assess barriers to children’s active play in nature, but the parent sample size was small (*n* = 14 parents), and the overall aim of the study was to inform pediatric health care providers and advocate for active outdoor play in a clinical setting [34]. The last study focused on generational differences in outdoor play between mother/child dyads in a sample of 830 mothers, but the study was published in 2004, meaning it may not be as informative to current barriers to outdoor time in early childhood [32]. Collectively, these studies identified the following primary barriers to children’s outdoor time: (1) safety concerns [32,34]; (2) screen time and other digital media [32]; (3) lack of adult supervision [32]; (4) lack of physical and financial resources (e.g., access to safe outdoor spaces, time, etc.) [34]; and (5) weather concerns [34]. Despite this important work, mixed methods research aimed at identifying general barriers to and facilitators for outdoor time in childhood, specifically in communities of color, is absent.

Community-based participatory research (CBPR) approaches are critical to an equity-centered research framework [35,36,37,38,39]. CBPR typically uses a multi-step approach to disable the traditional hierarchical structure of research, including collaborating with community partners as equal partners and leaders, generating research questions that align with the needs of the community, and sharing decision-making in all phases of the research process including interpretation of results and authorship roles and order [40,41]. Historically, Eurocentric research approaches and results do not align or translate, respectively, to populations with different cultural frameworks and backgrounds. Research is accelerating in this area, particularly in adult populations [42,43], but research in culturally diverse youth highlights the importance of assessing the needs of communities and youth of color [44,45,46] to better identify solutions to improve health in these populations. The National Institute on Minority Health and Health Disparities (NIMHD) research framework [47] complements the CPBR approach by categorizing levels (e.g., individual, community, etc.) and domains of influence (e.g., biological, physical/built environment, etc.) on health outcomes. Given the importance of incorporating the voices of communities of color and the relative absence of data about outdoor time in youth in general, the CBPR and NIMHD frameworks informed all stages of this research project. Thus, the objectives of this research were to: (1) identify challenges and solutions to outdoor time for children; (2) assess the importance of outdoor time for children; and (3) evaluate results stratified by race/ethnicity.

## 2. Materials and Methods

### 2.1. Theoretical Framework

To systematically analyze factors influencing health disparities related to the relationship between the physical/built environment (availability and access to community outdoor resources) and the sociocultural environment (social networks; family/peer, community, and social norms; interpersonal, local, and structural discrimination), this study was theoretically guided by the research framework developed by the NIMHD [47]. To align with an equity-based approach, we embraced the CBPR framework to prioritize key stakeholder voices, perspectives, and needs in determining the challenges to and facilitators for outdoor time for children [44]. The mixed methods approach taken by the investigators in the current project is the first phase in the overall CBPR process. The second phase will be summarizing and delivering the results herein to a community-based working group tasked with designing a novel, community-informed intervention. The goal of this intervention will be to address and ameliorate some of the identified challenges herein, with the goal of increasing outdoor time for families and children and improving physical, behavioral, and mental health outcomes in communities of color. These research frameworks guided the definition of the overall research questions and the specific focus group questions.

We use the term people of color (POC) for participants who self-identified as African American/Black, American Indian/Alaska Native, Asian, Native Hawaiian, Hispanic, Mexican, or being of Latino/Latina ethnicity. The decision to use this term was made after careful consideration of the following: (1) from a conversation with our community partner, it is the term that best describes and is preferred by the community, and (2) this term was the most used by our research participants who self-identified as African American/Black, American Indian/Alaska Native, Asian, Native Hawaiian, Hispanic, Mexican, or being of Latino/Latina ethnicity. We acknowledge that this term is not universally accepted because it generalizes the vastly different historical and contemporary experiences of the people who self-identify or are identified as non-White. However, the small sample size of our study does not allow us to use the specific cultural, racial, and ethnic preferences of individual participants.

### 2.2. Setting and Participants

The Washington State University Institutional Review Board approved this study, and all participants provided informed consent. Eligibility criteria included families with a child enrolled at an outdoor preschool, being an educator or leader in an early childhood educational setting or being a parent in the community of interest. Participants were recruited between January and April 2022 through study flyers distributed by email and direct communication with the local early childhood education community network in King County, Washington. All participants received a USD 50 gift card for participating.

### 2.3. Procedures

Using a convergent parallel mixed methods design, qualitative focus groups (*n* = 50) and quantitative questionnaires (*n* = 49) were utilized to identify areas of convergence and divergence between the results. Fourteen focus groups were conducted between April and June 2022. Five focus groups were conducted with families with a child enrolled at an outdoor preschool, five with educators or leaders in an early childhood educational setting, and four with parents in the community of interest. All focus groups lasted approximately 1 h. Three focus groups were conducted in person, and eleven were conducted virtually over Zoom. Focus group sizes ranged from 3–10, though one focus group had a single participant due to last minute scheduling conflicts with other participants. Sessions were audio recorded, and participants were anonymized during the entire interview process. Participants completed a self-report questionnaire after finishing their focus groups. All questionnaires were administered using Research Electronic Data Capture (REDCap) [48,49], an electronic data capture tool hosted at Washington State University and took approximately 20 min to complete.

### 2.4. Measures

#### 2.4.1. Qualitative

The focus group moderators (the first and last authors) used a systematic and comprehensive protocol to conduct all focus group interviews. This process included a standardized focus group guide with open-ended questions and follow-up probes to examine challenges and solutions to outdoor time for children [50,51]. Challenges included the following: general (e.g., what makes it hard for your child to spend time outside in your community?), weather and access to appropriate outdoor attire, general access to outdoor spaces, safety concerns, and cultural norms and traditions associated with outdoor time. One question was asked regarding how much time children spend or should spend outdoors. The final question focused on identifying potential solutions to increase outdoor time for children.

#### 2.4.2. Quantitative

All survey data were collected using REDCap, an electronic database for secure data collection and storage. Demographic variables included age, sex assigned at birth, gender and racial identification, current relationship status (e.g., married/partnered, single, etc.), employment status, highest terminal degree (e.g., high school, bachelor’s degree, etc.), and annual household income (e.g., <USD 50,000, USD 50,000–89,999, ≥USD 90,000). Challenges to outdoor time were assessed using an adapted National Institutes of Health PhenX Toolkit instrument, Coping with COVID through Nature (CCN). The CCN was originally intended to assess pre-COVID and current nature exposure. For the purposes of this project, only questions about current nature exposure we asked. The CCN questions assess access to different natural settings (e.g., forested parks, trails, etc.) using 5-point Likert scale response options (strongly agree, agree, neutral, disagree, strongly disagree). Given there are no validated surveys assessing the perceived importance of outdoor time for health outcomes, questions regarding perceived importance of outdoor time, including cultural norms and traditions about outdoor time, were generated by consensus between the study team and community partners with 5-point Likert scale response options identical to the CCN. Finally, the study team generated questions regarding how much time an individual would like to spend outside each day (e.g., ideal outdoor time) and how much time they actually spend outside each day (e.g., actual outdoor time), with specific question for weekdays and weekends.

### 2.5. Data Analysis

#### 2.5.1. Qualitative

The transcription service Rev.com was used to transcribe the focus group audio recordings [52]. Transcripts were uploaded into Dedoose (version 9), a mixed methods data analysis software program for data management [53]. The qualitative analysts—the first and second authors—developed a detailed understanding of the data through multiple readings of the first three transcripts and a close examination of the themes repeated by the participants across the transcripts. Next, following methods of inductive and deductive thematic analysis [50], the qualitative analysts developed a codebook capturing and systematically organizing the main themes and subthemes that the participants discussed. While most of the main themes were defined deductively and aligned with the research questions, the minor themes were captured inductively and added under the main themes. Next, focus group transcripts were coded line-by-line by each analyst. When thematic saturation was reached and no new codes emerged, the codebook was finalized. The final codebook consisted of themes, subthemes, operational definitions, and examples of each theme. The analysts reviewed all the transcripts in duplicate to be sure the final codes were applied across all transcripts. The qualitative analysts met bi-weekly to clarify and compare themes, discuss coding progress, and share monthly updates with the research team. The inter-rater reliability was 84% of Cohen’s kappa value score, suggesting excellent agreement [54].

#### 2.5.2. Quantitative

Descriptive statistics were calculated using means and standard deviations for continuous variables, and frequencies or counts for categorical variables. Results are presented stratified by self-identification of POC—Native Hawaiian and/or African American/Black and/or American Indian/Alaska Native and/or Asian and/or being of Hispanic, Mexican, or Latino/Latina ethnicity compared to those who identify as Caucasian/White. All analyses were conducted using Stata 17 [55].

#### 2.5.3. Mixed Methods

After qualitative and quantitative data were analyzed, the two data sets were compared following mixed methods research approaches, including completing a detailed assessment of concordant and discordant concepts among the findings [56]. Using convergent parallel design, a joint display table was created to better understand how the results confirmed, disconfirmed, or expanded on each other [56]. This mixed methods approach provided deeper insights into the main themes of the research. Quantitative results expanded qualitative results by providing insights into: (1) stratified differences between POC and White participants, and (2) numerical values that provided a greater understanding of the magnitude of the results. Qualitative results expanded quantitative results by exploring the rationale and gaining a deeper understanding of the perceptions, importance, and challenges of spending time outdoors.

## 3. Results

Nearly all the participants completed a survey following focus group completion (98%). Twelve participants self-identified as POC (24%), 34 as White (69%), and three did not specify race or ethnicity (6%). One participant who identified as POC did not respond to any other survey items and was excluded from quantitative analyses for a final sample size of *n* = 45. The study sample was overwhelmingly female (93%), and the mean age was 34 years old. In comparison to their White counterparts, POC participants were more often married or partnered (73% vs. 65%), not currently employed (36% vs. 15%), had high school/GED as the highest terminal degree (36% vs. 9%), and annual household income < USD 90,000 (64% vs. 56%)—see Table 1.

In the focus group interviews, the participants discussed five main themes: (1) challenges for children’s outdoor time; (2) solutions for children’s outdoor time; (3) perceived importance of outdoor time; (4) cultural values and perspectives toward outdoor time; and (5) the influence of the built environment on children’s outdoor time (e.g., household and neighborhood). While the first four themes were defined deductively, the last one was defined inductively. Since the qualitative and quantitative data were cohesive, the results from both data sets were simultaneously integrated to provide comprehensive and validated results [56]. Quantitative and qualitative findings are congruent throughout all the main themes (see Table 2).

### 3.1. Challenges to Outdoor Time

#### 3.1.1. Safety, Weather, and Financial Challenges

The most discussed challenges to outdoor time were safety and weather. Based on the survey, 27% of POC participants and 3% of White participants disagreed with the statement, “I feel safe in outdoor spaces in my community” (see Table 3). Participants often mentioned their worries regarding the danger of people in public parks (e.g., individuals with mental health challenges, people having their dogs unleashed) and the danger of the place itself (e.g., hypodermic needles left on playgrounds, slippery surfaces where their children might get injured, branches falling in a storm). POC participants reported feeling unsafe in outdoor spaces, either due to their own experiences or generational perceptions of safety and belonging in outdoor spaces, which were sometimes perceived as White spaces. Differences in perception of safety varied by generation and urban versus rural settings. Many participants saw unstructured outdoor time in a rural setting as safer than outdoor time in urban areas. Lastly, participants mentioned concerns about safety from extreme weather, climate change’s impacts on weather, and subsequently children’s outdoor time.

While inclement weather was the primary challenge to outdoor time discussed in most focus groups, most participants believed that proper clothes and gear would address this concern. The larger Seattle metropolitan area, with its mild climate, was therefore perceived as comfortable and safe to spend time outdoors, as opposed to the extremely hot weather in other areas of the US. Most participants said kids tend not to mind cold and wet weather, but their caretakers did not feel comfortable outside in the rain or cold weather. Three main barriers related to gear were discussed: financial burden (quality gear is expensive and extended daily outdoor time requires multiple versions of the same item); time burden on parents (gear needs to be cleaned regularly); and hardship of selecting adequate gear, especially for kids with different sensory needs. The survey results reflected these findings, with 45% of POC and 15% of White participants saying they did not have all the gear needed to do outdoor activities (see Table 3). Financial challenges related to the outdoors were brought up by a few participants, mostly linked to equipment for winter sports. A few participants talked about growing up in financial adversity where their caretakers were focused on working to provide basic needs for the family. Given these barriers, outdoor time was unapproachable, regardless of knowledge of the benefits of outdoor time. Participants in three focus groups discussed a theme of privilege while talking about challenges to outdoor time. The privilege was either interpreted in relation to economic status or racial identity. The common theme was that when faced with obstacles that impede children and families from spending time outdoors, those with privilege can more easily overcome these barriers, which increases their access to the outdoors.

#### 3.1.2. Access to Nature and Outdoor Spaces

Access to the outdoors was primarily discussed in relation to distance and time. Some participants said parks were too far for them to walk to if they did not have a car, or the roads were unsafe for kids to bike or walk to green outdoor spaces. Participants shared concerns about decreasing wilderness and green spaces within metropolitan areas. Quantitative data further exposed differences in outdoor accessibility between POC and White participants and in relation to community outdoor resources. Outdoor spaces were more easily accessible for White participants than for POC participants—18% of POC participants disagreed with the statement, “It is easy to access outdoor spaces in my community”, while only 3% of White participants disagreed with this statement (see Table 3). Easy accessibility to resources for outdoor time included neighborhood parks (64%), home-based activities (49%), forested parks (33%), trails for hiking (24%), community/family gardens (20%), and water access for swimming (18%). Neighborhood parks and community/family gardens were more accessible for White participants than for POC participants (68% vs. 55%, 24% vs. 9%, respectively). On the contrary, trails for hiking and water access for swimming were more accessible to POC participants (36% vs. 21%, 27% vs. 18%, respectively, see Table 3).

#### 3.1.3. Lack of Time and Competing Priorities

Lack of time and competing priorities presented a consistent barrier that parents experienced in their everyday life. Incongruent preferences between caretakers and children in terms of free time activities were mentioned by many parents as well; while their children would like to spend as much time as possible outside, parents have errands to run or would rather spend their free time differently. Participants often said they felt exhausted from juggling competing priorities, one of them being ensuring their kids spend time outdoors daily. They suggested this issue could be solved by increasing outdoor time during school hours. Based on the survey results, POC participants compared to their White counterparts more often reported it was hard to find time to be outside (36% vs. 18%, respectively, see Table 3). A minor subtheme was electronics and screen time negatively affecting children’s motivation for outdoor time.

#### 3.1.4. Lack of Parks and Playgrounds

Only a few participants mentioned a lack of parks and playgrounds in their neighborhoods as a barrier. More participants discussed parks needing to be equipped appropriately (e.g., too many kids for the available playground equipment, lack of playgrounds equipped for their kid’s age group, or lack of playgrounds protecting their children against rain). A few participants said that some playgrounds in their neighborhood were closed due to the COVID-19 pandemic, resulting in limited opportunities to spend time outdoors. While some parents saw this as an opportunity to reimagine outdoor time and discover new outdoor activities, others felt the COVID-19 pandemic led to decreased outdoor time, often connected to their worries about getting infected.

### 3.2. Solutions to Increase Outdoor Time

#### 3.2.1. Increasing Outdoor Time during the School Day

The most frequent solution to increase outdoor time for children was working through schools. This topic came up in almost every focus group and was described in several ways. Some participants shared stories of their childhoods, where recess at school was the primary time they could play outdoors. Others discussed the role of schools in educating families about the importance of outdoor time for their children. There was a consensus that most schools do not offer enough time outside for children. Participants expressed frustration that many schools discouraged time outdoors if the weather was cold or rainy or simply did not offer sufficient recess time for children, especially in middle or high schools. The solutions described by participants included increasing recess time, supporting children going outdoors despite rainy or cold weather, and offering regular coursework in an outdoor environment. Participants expressed that increasing the amount of outdoor time available for children would improve their attention span, cognitive abilities, learning, and social experiences.

#### 3.2.2. Increasing Access and Availability to Outdoor Schools

Outdoor schools were also discussed as a solution to provide children with more opportunities to spend time outdoors. This was mainly discussed in the context of early childhood education. Many participants talked about the high demand for outdoor schools that led to long waitlists and expressed a desire for more options with a similar nature-based learning model. Outdoor educators discussed the possibility of raising awareness among early childhood educators of the importance of outdoor time and the nuances of teaching children in an outdoor environment.

#### 3.2.3. Organized, Accessible, and Diverse Outdoor Activities

Similarly, participants often talked about organized outdoor activities that would address the challenges outlined above, especially the lack of time parents have during weekdays to take their children outdoors. Participants envisioned the positive impacts of outdoor afterschool programs that could be offered for different age groups at community centers, public schools, private daycare centers, or informally as an agreement between multiple families to share the responsibility of supervising their children outdoors during consistent days per week. Several positive outcomes of children playing together outside were also mentioned, such as increased physical activity.

Participants also talked about the positive impact of diverse outdoor activities. They wished public schools and community centers engaged children through various outdoor activities, both structured (e.g., swimming, ball games, biking, and hiking) and free unstructured playtime in parks. The favorite activities for parents and their children included playing at a playground, jumping in puddles, biking, walking around the neighborhood, camping, hiking, exploring the woods or the beach, playing soccer, swimming, and jumping on a trampoline.

#### 3.2.4. Community Networks, Communication, and Raising Awareness about Outdoor Time

Some participants described the strong community networks they grew up in as an important aspect of increased outdoor time. They shared childhood memories of being free to run around with other kids from their community while “all the street was watching”. Contrary to their childhood experience, they said it was rare to find the same social cohesion nowadays in their neighborhoods, which were characterized by frequent turnover of residents. The sparse community life in the participants’ neighborhoods was itself a major barrier to outdoor time for their kids, because it is directly linked to adult (supervision) time, which was defined as very limited. Creating and strengthening communities was therefore seen as a promising opportunity for interventions leading to increased outdoor time in early childhood.

Participants mentioned the importance of communicating and sharing information about the benefits of outdoor time for children, such as locally organized activities, and how to overcome inclement weather on media sources frequently accessed by parents (e.g., social media, radio stations, and television). This closely relates to another critical facilitator to increase outdoor time in childhood—raising awareness about the importance of outdoor time. Participants expressed multiple times that parents should be better informed about the value of outdoor time and its impact on their children’s physical, mental, and behavioral development.

#### 3.2.5. Inclusivity, Safety, and Access

In a focus group with participants who all self-identified as POC, the need to support POC’s advocacy for increased outdoor time was heavily discussed. Participants expressed the need to better inform POC communities about the importance and positive outcomes of children’s outdoor time, such as through social media, because considering other competing priorities that their communities face daily, outdoor time is often perceived as a luxury and not a key lifestyle behavior to promote health. Another topic discussed was the emphasis placed on children’s academic achievement by parents and in school, which results in outdoor time being a lower priority. According to participants who were preschool teachers (but not exclusively), schools should embrace and promote the idea of authentic nature-based learning versus solely focusing on academics.

Increasing safety was seen as another way to encourage outdoor time. Participants talked about the importance of cleaning parks of hypodermic needles and having security personnel in parks. This was extremely important especially for families of color who might otherwise not feel safe in the city parks. Others discussed the importance of increasing access to parks by building more sidewalks in the vicinity of parks. Creating new parks, even very small ones, while dividing them from roads by a fence was also seen as a major promoter for daily outdoor time. A few participants mentioned free and/or accessible gear and equipment as a facilitator as well. They suggested there should be easily accessible information about which clothes are proper for rainy and cold weather, and events should be organized where outdoor clothes and gear can be exchanged.

### 3.3. Perceived Importance of Outdoor Time

Participants in all focus groups were asked how much time children should spend outside. Answers varied widely, from 2 h to “as much as possible”. Other participants expressed that it depends on the child and their individual needs, but there was overall unanimity that children should be spending time outdoors. The positive outcomes of outdoor time were seen as improving both physical and mental health. This aligned with the quantitative results given that: (1) 100% of the participants agreed in the survey that outdoor time promotes health and wellness; (2) 100% of participants agreed in the survey they were more physically active when spending time outdoors; (3) 100% of POC and 94% of White participants agreed that their physical health was better when spending time outside; and (4) 100% of POC and 97% of White participants agreed that their mental health was better when spending time outside (see Table 4). All groups reported spending more time outdoors on weekends. While both groups reported they would ideally get ~6.5 h of outdoor time on weekends, POC and White participants reported they spent ~2.5 h and ~3.5 h per day outside on weekend days, respectively (see Table 4).

### 3.4. Cultural Norms and Traditions toward Outdoor Time

Participants were encouraged to reflect on how their cultural norms and traditions influenced their perspective on outdoor time for children. POC participants tended to define the concept of culture based on their ethnic or racial identification, and they mainly talked about cultural traditions and values typical for their ethnic group. Other participants talked about culture in terms of their family traditions. This question often made participants reflect on their childhood and the environment in which they grew up. Many mentioned that their caregivers had no safety concerns, usually due to strong community networks. Others shared that outdoor time is perceived very differently by families who were outdoor laborers, refugees, or had experienced trauma outdoors, either personally or ancestrally. These factors impacted their perceived importance of outdoor time and how likely they were to engage in outdoor activities. POC participants who were second-generation immigrants expressed multiple times that they were often confronted with a lack of understanding from their parents or relatives when they expressed wanting to spend extended time outdoors with their children. They talked about their parents associating outside activities with hard work and negative connotations such as “dirty” or “cold”, and their parents struggled to understand the desire to be exposed to such an environment despite it not being necessary. A discomfort with being “dirty” after playing outdoors was thought to be more prevalent among POC communities.

The survey data align with these cultural experiences and perspectives on outdoor time expressed by focus group participants. While 76% of White participants agreed with the statement, “a lot of people in my culture spend time outside”, only 18% of POC participants agreed with this statement. More White participants (79%) believed outdoor time was an important value in their culture than POC participants (50%; see Table 5).

Another subtheme frequently discussed through the focus groups was how culture shapes everyone’s relation to nature, subsequently guiding their perceptions and activities practiced outdoors. A striking contrast was discussed between White Americans’ perceptions of nature, sometimes described in terms of conquering nature, and perceptions of other cultures and ethnic groups, which was described as an intimate relation to nature that is grounded in everyday connection to the natural world.

### 3.5. Home, Neighborhood, Built Environment, and Children’s Outdoor Time

In response to the question about challenges to outdoor time, some participants shared how their home environment shaped their access to the outdoors. Participants living in apartment buildings without direct access to the outdoors talked about extra layers of difficulties they experience when they want their children to spend time outside, for the most part linked to the caretakers’ time involvement and other aspects of accessibility (see Section 3.1. Challenges to Outdoor Time). Since apartment buildings tend to be in densely populated urban environments, even leaving the apartment building with their children raised specific safety concerns. One participant talked of their apartment building’s parking lot being the only easily accessible outdoor space in contrast to the yards typical of family houses. It was also discussed that while having a yard provides an opportunity to access an outdoor space quickly, parents did not always feel fully comfortable letting their children play there without adult supervision. Yards might provide their children with extra time daily to spend outdoors and engage in physical activities (digging holes, running around, etc.), but participants described a meaningful contrast between playing in the yard versus making a concentrated effort to spend outdoor time in a park. Different characteristics of the urban outdoor environment (e.g., playground on a concrete lot vs. playing in a park vs. hiking in the wilderness) and the effects these environments have on children’s health were part of the conversation. The answers to the survey questions about the importance of outdoor community resources provide insight into the environments that are most welcoming for outdoor time. All POC and 94% of White participants found having neighborhood parks very important, 91% of POC and 82% of White participants found having forested parks in their city or state very important, 91% of POC and 76% of White participants found having access to home-based activities (playing outside) very important, 91% of POC and 68% of White participants found having trails for hiking very important, 80% of POC and 59% of White participants found having water access for swimming very important, and 73% of POC and 56% of White participants found access to community/family gardens very important (see Table 6). Some participants also mentioned differences between rural and urban environments as one of the guiding principles influencing outdoor time. Participants talked about their childhood experiences of growing up in rural areas where all the world behind their door was essentially a playground and where their caretakers did not need to worry about the safety aspects listed above that are an essential part of daily life in an urban environment.

## 4. Discussion

This study is the first to use a mixed methods approach to identify challenges and solutions to outdoor time for children, to assess the importance of outdoor time for children, and to utilize an equity-based lens to evaluate these results stratified by race/ethnicity. The main challenges identified which limit outdoor time for children were safety concerns, inclement weather/lack of appropriate gear, limited access to outdoor spaces, and parents’ work schedules. POC consistently reported facing greater obstacles to identified barriers compared to their White counterparts. Proposed solutions included increasing outdoor time through the school system, outdoor schools, organized activities, and community networks. All participants reported that outdoor time was important, that it promoted health and wellness, that they were more physically active when they spent time outside, and that outdoor time improved both physical and mental health.

Our findings align with previous research [32,34] that identified the following barriers to outdoor time: safety concerns, inclement weather/lack of gear, lack of access to outdoor spaces, and parents’ lack of time (e.g., work schedules). Our findings extend the previous works in five noteworthy ways. First, we used a rigorous mixed methods approach and stratified the results by race/ethnicity, whereas the previous work was either exclusively qualitative [34] or quantitative [32] and did not include stratification by race/ethnicity. Second, not all our respondents saw inclement weather as negatively affecting outdoor activities; some perceived it as a vital resilience builder and an opportunity to explore new outdoor activities. Third, public outdoor spaces often do not feel safe for POC. Consistently shared experiences from POC participants highlighted incomparable differences in what people fear in both local (e.g., city) and national parks based on their racial or ethnic identity and the color of their skin. While White participants were primarily concerned about hypodermic needles and strangers with mental health challenges, POC people talked about being afraid of violence and for their very lives while spending time in parks. This finding complements other research in adult populations indicating that public green spaces are perceived as a setting for violence for communities of color [23,57]. Fourth, we prioritized identifying solutions that would allow children to spend more time outside. Competing priorities and subsequent time constraints experienced by caretakers are key barriers to children’s outdoor time and were directly mirrored in the solution to promote outdoor time in school-based settings. Fifth, we assessed the perceived importance of outdoor time. While perhaps intuitive, it is critical when conducting CBPR to incorporate the perspectives of the priority population or community instead of making assumptions about priorities and values regarding outdoor time for their children [37,38,40]. All participants reported that children should spend time outside. Most participants felt that the amount of daily outdoor time they wanted for their children was unachievable given the burden of everyday parenting responsibilities and work schedules. Participants suggested that school-based settings, where children spend most of their weekday time, present an opportunity to offer outdoor time during the school day. School-based built environment changes align with the NIMHD research framework as a critical physical/built environment domain of influence that provides the opportunity for researchers and policymakers to reduce health inequities [47].

Incorporating outdoor time into school-based built environments, either through recess, nature-based learning opportunities, or greening schoolyards, would address and positively influence the following barriers in three ways: (1) caregiver lack of time to prioritize outdoor time; (2) safety in outdoor settings; and (3) comfort with inclement weather. First, while participants reported that residential built environment, specifically differences in living in apartment buildings versus in houses with yards, influenced daily outdoor time, there was a consistent theme that all parents and caregivers felt overextended and that supporting outdoor time in a school-based setting was a priority. While the neighborhood and residential built environment typically influences how much daily outdoor time children get on weekends, a school-based solution would likely have a larger impact given that most children attend school on weekdays which represents a larger proportion of their total time (weekly and yearly). Thus, school-based built environments are the intuitive target for interventions for equity-centered approaches that reduce health inequities in childhood. Second, schools provide safe (and monitored) outdoor spaces for all children regardless of ethnic and racial background, therefore increasing safe and welcoming outdoor time opportunities for all children. Third, schools can engage in encouraging a value system that normalizes outdoor activities, develops habits to stay comfortable in rainy and cold weather, and provides proper gear to enjoy outdoor time year-round for all children, including families that could not otherwise afford it.

In addition to providing a solution to the identified barriers to outdoor time for children, an outdoor-oriented approach in school settings would further support the numerous health benefits of nature time for all children, regardless of racial or ethnic identity [1,2,3,4,6,7,9,11,13,15,16]. School settings have historically offered tremendous opportunities to support academic [58], developmental [59,60], physical [58,61], and nutritional health [62,63,64] in childhood. Given this commitment to the health of the whole child, improving access to nature-rich outdoor play and learning spaces is a straightforward next step to reduce health inequities and optimize health in childhood. Greening schoolyards—the process of transforming a traditional asphalt-rich play space into a nature-rich and engaging play space—shows tremendous promise for improving numerous health outcomes in childhood and adolescence [59,60,65]. Increased access to green spaces during the school day provides an immense opportunity to narrow health inequities experienced by families of color by providing identical opportunities to health-promoting spaces and activities.

The vast majority of the studies that explore outdoor time from the perspectives of POC focus on promoting physical activity, not outdoor time, in adult populations [22,26,28,29,66,67,68,69]. Our study is unique in prioritizing outdoor time specifically in childhood and evaluating differences for participants that identify as POC and White. We found that cultural values and traditions influenced numerous aspects of outdoor time in childhood, including the perceived importance of nature time for children, access, feelings of safety, and outdoor activities. While 50% of our POC participants agreed that outdoor time is an important value in their culture, only 18% believed that a lot of people in their culture spend time outside. Thus, there is a large discrepancy between POC valuing outdoor time and spending time outside. This discrepancy was not present for White participants. Furthermore, while 100% of POC and White participants considered outdoor time important, our quantitative data showed that the amount of time these groups spend outside differs and outdoor spaces are not equitably accessible to them (Table 4). These trends are most likely influenced by the barriers outlined above (e.g., safety) that are experienced by POC and White participants differently. In addition, outdoor spaces (e.g., neighborhood and forested parks, home-based activities—playing outside, and community or family gardens) were broadly more accessible for White participants than for their POC counterparts (Table 3). This finding has been reported in previous literature in adult populations and is usually explained by residential and other forms of structural discrimination [31,70,71,72,73]. Interestingly, POC participants in our study had better access to hiking trails than their White counterparts. This finding may be capturing historical racial segregation and ongoing gentrification situating POC communities to the city periphery [74,75], and may position POC communities in the Seattle area in closer proximity to nature. However, given the serious safety concerns that POC participants face, the essential question of usability despite accessibility remains to be answered and merits future research. Such insights are crucial to consider for future research aimed at developing culturally sensitive and tailored interventions in settings characterized by racially and ethnically diverse populations.

This study has several limitations. First, the sample size is limited (*n* = 45); therefore, our analyses are descriptive rather than inferential and the results should be interpreted accordingly. Since only 11 (24%) of our participants self-identified as POC, the results of this study should be considered exploratory and hypothesis-generating in nature. Future research with a larger sample size could provide inferential results with broader generalizability. Second, our survey did not include questions about solutions to outdoor time, which impeded mixed method analyses of this topic across both data sets. Third, the study was conducted in the Seattle metropolitan area from April to June 2022. This time of year often correlates with the end of the long rainy season, which may have biased responses about concerns regarding inclement weather. Fourth, no validated instruments were available for the assessment of the quantitative outcomes. Given research regarding outdoor time and nature exposure is relatively new, it is understandable that reliability and validity testing of instruments has not been done. Prioritizing psychometric testing for instruments evaluating challenges to outdoor time and perspectives about how outdoor time influences health is critically needed.

There are also numerous strengths to this study. First, the enrollment of the participants in this study was supported by our community partner, which is the largest exclusively outdoor preschool in the US. The community partner’s support and research engagement enabled us to collect rich experiences and valuable insights into the topic from outdoor educators. Second, this work was informed by the historical and systemic lack of representation of young children and families of color in nature in the Seattle area, and the accumulating research that outdoor time improves physical health, development, and mental health outcomes in childhood. Third, our mixed method study brings innovative findings into the experiences and perspectives of POC families with young children who are underrepresented in early childhood outdoor education research.

## 5. Conclusions

The main challenges to outdoor time for children were safety concerns, inclement weather, parents’ work schedules and competing priorities related to childcare, and lack of access to outdoor spaces. Challenges to and solutions for children’s outdoor time were similar between POC and White participants; however, they were experienced in greater magnitude by POC participants, especially regarding safety concerns and access to community outdoor resources. All participants considered daily outdoor time in childhood highly important despite different cultural backgrounds remarkably shaping their perceptions of outdoor time, the safety of outdoor settings, and preferred outdoor activities. There was overall unanimity that outdoor time should be promoted and increased in school-based settings. Our findings contribute to an ongoing discussion of nature as a social determinant of health and a call for creating safe, racism-free outdoor spaces embraced by school programs as an intuitive way to promote health equity for all children. Insights from this study may inform future research efforts aimed at culturally appropriate interventions to increase outdoor time in childhood that would ultimately advance health equity in marginalized pediatric populations.

## Figures and Tables

**Table 1 ijerph-20-07149-t001:** Demographic characteristics among the participants (*n* = 45) ^1^.

	Race			
	POC (*n* = 11)		White (*n* = 34)	
**Age, *mean years (SD)***	35	(5)	33	(7)
**Female sex assigned at birth, *n (%)***	11	(100%)	31	(91%)
**Currently married or partnered, *n (%)***	8	(73%)	22	(65%)
**Currently employed, *n (%)***	7	(64%)	29	(85%)
**Completed education, *n (%)***				
High school/GED	4	(36%)	3	(9%)
Technical/vocational, associate, or bachelor’s degree	3	(27%)	17	(50%)
Post-graduate or professional degree	4	(36%)	14	(41%)
**Annual household income, *n (%)***				
<USD 50,000	4	(36%)	11	(32%)
USD 50,000–89,999	3	(27%)	8	(24%)
USD 90,000 or more	4	(36%)	15	(44%)

^1^ *n* = 3 participants missing race, *n* = 1 participant missing all survey items besides race.

**Table 2 ijerph-20-07149-t002:** Mixed methods joint display table results.

Main Topics	Quantitative Results	Qualitative Results	Mixed Methods Comparison
**Identified Barriers to Outdoor Time**			
**General**	Safety, lack of time, lack of gear, access (see statistics in rows below).	Safety, weather, time, competing priorities, gear and other financial challenges, access, electronics, destruction of natural places.	Confirmation and expansion While quantitative data provide insights into four barriers to outdoors, the qualitative data explore more barriers and capture why each barrier is a challenge to outdoor time. Quantitative data are stratified by race, therefore providing important insights into how the racial identity of the participants influences barriers to outdoors.
**Safety**	A total of 27% of POC participants and 3% of White participants disagreed with the statement, “I feel safe in outdoor spaces in my community”.	Safety was one of the main barriers to outdoor time. Participants mentioned worries regarding the danger of people in public parks, the danger of the place itself, and safety concerns for POC in outdoor spaces. Differences in perception of safety also varied by generation and urban versus rural settings.	Confirmation and expansion While the survey question asked participants about their own feelings of safety in outdoor spaces in their communities, during focus groups, the participants talked mainly about safety for their children. POC participants shared feeling unsafe in public green spaces across both data sets. Qualitative findings further expanded multiple layers and reasoning of feelings of unsafety.
**Lack of gear and/or financial challenges**	A total of 45% of POC and 15% of White participants did not have all the gear needed to do outdoor activities.	Gear as a barrier to outdoors was mentioned by participants mostly in relation to inclement weather. Three main barriers related to gear were discussed: financial burden, time burden on parents, and hardship of selecting adequate gear.	Confirmation and expansion Quantitative and qualitative results align. Qualitative results provide insight into the rationale (e.g., financial burden) behind gear being a barrier to the outdoors.
**Access**	Outdoor spaces were easier to access for White vs. POC participants—3% of White participants vs. 18% of POC disagreed with the statement, “It is easy to access outdoor spaces in my community”. For detailed results regarding access to community outdoor resources.	Travel distance was the primary access barrier discussed. Some participants said parks are too far for them to walk to if they do not have a car, or the roads are unsafe for kids to bike or walk. Only a few participants mentioned a lack of parks and playgrounds in their neighborhoods as a barrier. Participants more often discussed parks needing to be properly equipped.	Confirmation and expansion Quantitative and qualitative results align. Both provide different insights into outdoor accessibility. Quantitative data expose differences in outdoor accessibility between POC and White participants and in relation to community outdoor resources. Qualitative data provide insights into accessibility to more outdoor spaces than listed in a survey and into difficulties with access in the context of other challenges participants (especially parents) face while wanting daily outdoor time for their children.
**Lack of time**	A total of 36% of POC participants and 18% of White participants reported finding time to spend outside was hard.	Participants often talked about competing priorities parents are experiencing in their everyday life that result in a lack of time for the outdoors.	Confirmation and expansion Quantitative and qualitative results align. Qualitative results bring insights into the rationale (e.g., competing priorities) behind the lack of time being a barrier to the outdoors.
**Identified Solutions to Outdoor Time**			
**Solutions to outdoor time**	Not asked.	Solutions to outdoor time often mirrored the discussed challenges. The most proposed solutions included: (1) incorporating outdoor time in school settings; (2) increasing the availability of outdoor schools, organized activities, community networks, and diversity of outdoor activities; (3) raising awareness about the importance of outdoor time; and (4) increasing safety in parks, access to parks, and free or accessible gear and equipment.	Not Applicable We did not include questions about solutions to outdoor time in the questionnaire due to the explorative character of our research which caused an inability to use a mixed methods comparison of this theme.
** Other Influences on Outdoor Time**			
**Ideal vs. actual amount of outdoor time**	POC and White participants reported they would *ideally* get ~3 h and ~4 h of outdoor time daily, respectively. POC and White participants reported they *actually* get ~2 h and ~3 h of outdoor time daily, respectively.	There was overall unanimity that children should be spending time outdoors. Opinions on how much time children should spend outside varied widely, from 2 h to “as much as possible”, based on the child and individual needs. Although all the participants agreed that outdoor time was beneficial for children, the overwhelming majority said that children nowadays do not routinely spend enough time outdoors. For some participants, this included their own children.	Confirmation and expansion Outdoor time was found important among all participants and across both data sets. Quantitative findings show differences between ideal vs. actual hours of daily outdoor time. Qualitative results provide more nuanced insights into the ideal amount of outdoor time and whether it is achievable.
**Outdoor time and mental and physical health**	It was found that 100% of participants agreed that outdoor time promotes health and wellness. A total of 100% of POC and 94% of White participants agreed that their physical health was better when spending time outside. A total of 100% of POC and 97% of White participants agreed that their mental health was better when spending time outside.	Participants talked about outdoor time improving children’s physical health by being physically active, burning off energy, and improving tolerance to allergens. They also talked about the positive impacts of outdoor time on their own and their children’s mental health by promoting learning, encouraging independence, gaining emotional intelligence, overcoming obstacles, dealing with discomfort, and bringing joy.	Confirmation Outdoor time was found to have positive impacts on mental and physical health and well-being across both data sets.
**Cultural perspectives and traditions on outdoor time**	A total of 50% of POC and 79% of White participants believed that outdoor time was an important value in their culture. A total of 18% of POC and 76% of White participants agreed that a lot of people in their culture spend time outside.	Perceptions of nature and outdoor time were influenced by cultural values and family traditions of the participants. Participants in a focus group where everyone self-identified as a POC discussed the need to better inform POC communities about the importance and positive outcomes of children’s outdoor time, such as through social media, because considering other competing priorities their communities face daily (see barriers to outdoor time), outdoor time is often perceived as a luxury and not a necessity.	Confirmation and expansion Quantitative and qualitative results align. Each data set provides a deeper understanding of cultural distinctions in experiences and perspectives of the outdoors.
**Built environment and children’s outdoor time**	For 100% of POC and 94% of White participants, neighborhood parks were a very important community resource. Home-based activity, such as playing outside, was very important for 91% of POC and 76% of White participants. For detailed results regarding the importance of community outdoor resources.	Many participants shared how their neighborhood environment shaped outdoor time for their children or children in their community. Participants living in apartment buildings without direct access to the outdoors talked about additional challenges in accessing outdoor spaces. Different characteristics of the urban environment and the effects these environments have on children’s health were discussed. Favorite activities for parents and their children included playing at a playground, jumping in puddles, biking, walking around the neighborhood, camping, hiking, exploring in the woods or the beach, playing soccer, swimming, and jumping on a trampoline.	Confirmation and expansion Quantitative and qualitative results align. The quantitative questions about the importance of outdoor community resources revealed that outdoor spaces are very important for both POC and White participants. A higher proportion of POC participants indicated outdoor spaces were very important across all outdoor space domains compared to White participants. Qualitative findings provided insights into popular activities that parents like to do outdoors with their children. The qualitative results also convey perspectives about the impacts of different characteristics of the built environment on outdoor time.

**Table 3 ijerph-20-07149-t003:** Challenges to outdoor time (*n* = 45) ^1^.

	Race			
	POC (*n* = 11)		White (*n* = 34)	
** *Safety and weather* **				
**I feel safe in outdoor spaces in my community, *n (%)***				
Strongly disagree/Disagree	3	(27%)	1	(3%)
Neutral	3	(27%)	12	(35%)
Strongly agree/Agree	5	(45%)	21	(62%)
**I do not have all the gear I need to do outdoor activities, *n (%)***				
Strongly disagree/Disagree	2	(18%)	23	(68%)
Neutral	4	(36%)	6	(18%)
Strongly agree/Agree	5	(45%)	5	(15%)
** *Access to nature and outdoor spaces* **				
**It is easy to access outdoor spaces in my community, *n (%)***				
Strongly disagree/Disagree	2	(18%)	1	(3%)
Neutral	2	(18%)	6	(18%)
Strongly agree/Agree	7	(64%)	27	(79%)
**Neighborhood parks, *n (%)***				
No access	0	(0%)	0	(0%)
Difficult to access	3	(27%)	0	(0%)
Somewhat easy to access	2	(18%)	11	(32%)
Easy to access	6	(55%)	23	(68%)
**Home-based activity (playing outside), *n (%)***				
No access	1	(9%)	0	(0%)
Difficult to access	2	(18%)	5	(15%)
Somewhat easy to access	3	(27%)	12	(35%)
Easy to access	5	(45%)	17	(50%)
**Forested parks in your city or state, *n (%)***				
No access	0	(0%)	0	(0%)
Difficult to access	2	(18%)	3	(9%)
Somewhat easy to access	6	(55%)	19	(56%)
Easy to access	3	(27%)	12	(35%)
**Trails for hiking, *n (%)***				
No access	0	(0%)	0	(0%)
Difficult to access	2	(18%)	6	(18%)
Somewhat easy to access	5	(45%)	21	(62%)
Easy to access	4	(36%)	7	(21%)
**Community/family garden, *n (%)***				
No access	1	(9%)	1	(3%)
Difficult to access	5	(45%)	5	(15%)
Somewhat easy to access	4	(36%)	20	(59%)
Easy to access	1	(9%)	8	(24%)
**Water access for swimming, *n (%)***				
No access	0	(0%)	1	(3%)
Difficult to access	5	(45%)	6	(18%)
Somewhat easy to access	3	(27%)	21	(62%)
Easy to access	3	(27%)	6	(18%)
** *Lack of time and competing priorities* **				
**It is hard for me to find time to be outside, *n (%)***				
Strongly disagree/Disagree	6	(55%)	20	(59%)
Neutral	1	(9%)	8	(24%)
Strongly agree/Agree	4	(36%)	6	(18%)

^1^ *n* = 3 participants missing race, *n* = 1 participant missing all survey items besides race.

**Table 4 ijerph-20-07149-t004:** Perceived importance of outdoor time among the participants (*n* = 45) ^1^.

	Race			
	POC (*n* = 11)		White (*n* = 34)	
**Outdoor time promotes health and wellness, *n (%)***				
Strongly disagree/Disagree	0	(0%)	0	(0%)
Neutral	0	(0%)	0	(0%)
Strongly agree/Agree	11	(100%)	34	(100%)
**I am more physically active when I spend time outside, *n (%)***				
Strongly disagree/Disagree	0	(0%)	0	(0%)
Neutral	0	(0%)	0	(0%)
Strongly agree/Agree	11	(100%)	34	(100%)
**My physical health is better when I spend time outside, *n (%)***				
Strongly disagree/Disagree	0	(0%)	0	(0%)
Neutral	0	(0%)	2	(6%)
Strongly agree/Agree	11	(100%)	32	(94%)
**My mental health is better when I spend time outside, *n (%)***				
Strongly disagree/Disagree	0	(0%)	0	(0%)
Neutral	0	(0%)	1	(3%)
Strongly agree/Agree	11	(100%)	33	(97%)
**Ideal outside time, *mean minutes (SD)***				
Typical weekday	177	(90)	264	(174)
Typical weekend	390	(156)	394	(238)
**Actual outside time, *mean minutes (SD)***				
Typical weekday	127	(150)	203	(143)
Typical weekend	161	(91)	217	(146)

^1^ *n* = 3 participants missing race, *n* = 1 participant missing all survey items besides race.

**Table 5 ijerph-20-07149-t005:** Cultural norms and traditions toward outdoor time among the participants (*n* = 45) ^1^.

	Race			
	POC (*n* = 11)		White (*n* = 34)	
**A lot of people in my culture spend time outside, *n (%)***				
Strongly disagree/Disagree	3	(27%)	1	(3%)
Neutral	6	(55%)	7	(21%)
Strongly agree/Agree	2	(18%)	26	(76%)
**Outdoor time is an important value in my culture, *n (%)***				
Strongly disagree/Disagree	3	(30%)	2	(6%)
Neutral	2	(20%)	5	(15%)
Strongly agree/Agree	5	(50%)	27	(79%)

^1^ *n* = 3 participants missing race, *n* = 1 participant missing all survey items besides race.

**Table 6 ijerph-20-07149-t006:** Home, neighborhood, built environment, and children’s outdoor time among the participants (*n* = 45) ^1^.

	Race			
	POC (*n* = 11)		White (*n* = 34)	
*Importance of resources for outdoor time for people in your community:*				
**Neighborhood parks**				
Not/Not very important	0	(0%)	0	(0%)
Somewhat important	0	(0%)	2	(6%)
Very important	11	(100%)	32	(94%)
**Forested parks in your city or state**				
Not/Not very important	0	(0%)	0	(0%)
Somewhat important	1	(9%)	6	(18%)
Very important	10	(91%)	28	(82%)
**Home-based activity (playing outside)**				
Not/Not very important	0	(0%)	1	(3%)
Somewhat important	1	(9%)	7	(21%)
Very important	10	(91%)	26	(76%)
**Trails for hiking**				
Not/Not very important	0	(0%)	0	(0%)
Somewhat important	1	(9%)	11	(32%)
Very important	10	(91%)	23	(68%)
**Water access for swimming**				
Not/Not very important	0	(0%)	3	(9%)
Somewhat important	2	(20%)	11	(32%)
Very important	8	(80%)	20	(59%)
**Community/family garden**				
Not/Not very important	0	(0%)	0	(0%)
Somewhat important	3	(27%)	15	(44%)
Very important	8	(73%)	19	(56%)

^1^ *n* = 3 participants missing race, *n * = 1 participant missing all survey items besides race.

## Data Availability

The data presented in this study cannot be shared through commonly used data sharing repositories. The consent did not address broad sharing of participants data, nor the potential risks associated with broad data sharing of these types of data.

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
