# Peer review of "Outdoor Time in Childhood: A Mixed Methods Approach to Identify Barriers and Opportunities for Intervention in a Racially and Ethnically Mixed Population"

_ijerph, 2023, doi:10.3390/ijerph20247149_

Round 1
Reviewer 1 Report (Previous Reviewer 3)
Comments and Suggestions for Authors
The updated version of the manuscript, labeled ijerph-2459662 and titled "Outdoor Time in Childhood: A Mixed Methods Approach to Identify Barriers and Opportunities for Intervention in a Racially and Ethnically Mixed Population," demonstrates greater coherence compared to its initial counterpart. The incorporation of additional details serves to elucidate the study's framework and the objectives pursued within the research. Despite the limitation posed by a relatively small sample size in the analyzed group, the manuscript maintains scientific rigor and offers a comprehensive analysis of the challenges encountered, including safety concerns, adverse weather conditions, parental work schedules, competing childcare priorities, and a lack of access to outdoor spaces.
The document not only identifies these challenges but also proposes viable solutions for enhancing outdoor time in childhood. It addresses issues such as safety concerns, inclement weather, parental work schedules, competing childcare priorities, and limited access to outdoor spaces. The robustness of the manuscript lies in its ability to delve into these intricacies and provide nuanced insights.
While acknowledging the constraint of a small sample size, the manuscript navigates this limitation skillfully, ensuring that its scientific integrity remains intact. The detailed exploration of challenges and proposed solutions contributes significantly to the field, offering valuable perspectives for future studies. The results, despite the limited sample, serve as a valuable foundation for subsequent research endeavors. Furthermore, the findings can be instrumental in initiating actionable steps for local institutions to improve both the quality and quantity of outdoor time in childhood for individuals across diverse racial and ethnic backgrounds. This inclusive approach recognizes the unique needs of people of color and white individuals, paving the way for equitable improvements in the realm of outdoor experiences during childhood.
The manuscript satisfies the requirements for being published.
Author Response
Please see the attachment.

Reviewer 2 Report (New Reviewer)
Comments and Suggestions for Authors
Without a doubt, the use of quantitative and qualitative methods strengthens the study presented, but the great weakness lies in the sample provided. Without a doubt, a larger sample would strengthen the conclusions.
I take this opportunity to suggest that the authors delve deeper into this line of research with more social descriptors, physical activity tests and greater sampling power.
Author Response
Please see the attachment.

Reviewer 3 Report (New Reviewer)
Comments and Suggestions for Authors
I did not read this article in the first review. This article is very interesting and well written. The only issue is that they used a quantitative questionnaire. It does not say anything about if it was validated, if it was developed for this population or anything. Where did it come from? Did they attempt to validate it? This is a glaring omission. The authors should address this before they publish this article as it will give them credibility.
If the questionnaire was not validated in any way then they need to add this to the limitations.
Otherwise this is a solid study.
Author Response
Please see the attachment.

This manuscript is a resubmission of an earlier submission. The following is a list of the peer review reports and author responses from that submission.
Round 1
Reviewer 1 Report
Comments and Suggestions for Authors
Thank you for the opportunity to review the manuscript "Outdoor Time in Childhood: A Mixed Methods Approach to Identify Barriers and Opportunities for Intervention in a Racially and Ethnically Mixed Population." While this topic is important and the collected data has the potential to be highly impactful for future readers, there are numerous concerns with the manuscript itself. Please see below for the three biggest concerns.
1) Persons/communities/families/etc. of color is not always the term of preference. While there are concerns about “BIPOC” as well, (I’d refer you to Deo, M. E. (2021). Why BIPOC fails. Va. L. Rev. Online, 107, 115.), it more commonly accepted in academic circles. Thus, I’d recommend the authors consider 1) can they be more specific about the communities in which their research concerns and 2) have a conversation about which term best describes said community. Additionally, the results use “BIPOC” while the introduction uses “persons of color,” please be consistent.
2) In the introduction, particularly the last paragraph, another sentence (or couple of sentences) are needed to fully address the “why” and specific literature gap.
3) Additionally, the results do not match the objectives. In the last paragraph of the introduction the authors state that the research will explore how outdoor time is experienced and approached based on ethnic & racial identity and sociocultural background – but the results only describe what barriers and facilitators exist – which are described in the introduction. Thus, the authors should either reframe their objectives (though this is not necessarily a better approach since the introduction clearly laid out that there has been significant amounts of work in this area) or rethink their results.
Comments on the Quality of English Language
The quality was high.
Reviewer 2 Report
Comments and Suggestions for Authors
Authors tackle an interesting topic which is the barriers and opportunities to perform outdoors activities regarding race and ethnicity. Authors developed a well-designed research method that provide a wide range of information. Nonetheless, the theoretical framework is feeble and some evidences of the differentiate qualitative analysis is required, before consider the manuscript for a further publication.
Therefore, it is necessary to extend the introduction including socioecological theories, school and community health promotion and social justice or race/ethnicity theories that provide some initial insights to the study topic.
Moreover, although the research method is well designed and developed, this reviewer would like to see the result of the inductive-qualitative analysis with, at least, the main categories of the study.
Reviewer 3 Report
Comments and Suggestions for Authors
The researchers examined obstacles and possibilities related to children’s outdoor time in a large metropolitan US area by comparing two groups from different racial and ethnic backgrounds. They also explored various factors that affect outdoor activities, including time availability, mental and physical well-being, cultural perspectives and traditions, and the surrounding physical environment.
This subject is significant as it sheds light on aspects that could impact children's health.
The introduction is well-written, pertinent to the field, and organized effectively, with appropriate references.
While the procedures are presented in detail and the qualitative and quantitative measures are clearly explained, the research results cannot be deemed statistically significant due to the limited size of the sample. The study included 11 participants who self-identified as BIPOC and 34 participants who identified as White, which may not be representative of a large US metropolitan area. Consequently, the findings may not accurately reflect the reality of the broader population. Even if the results are appropriately presented in relation to the sample size, and the discussions faithfully reflect the research outcomes, I strongly recommend increasing the sample size to enable a more accurate comparison between the two groups.
